# The eye, a spy hole on human mind: Spontaneous blink rate and amplitude, and their variability, as new psychobiological markers of anxiety

**Alon Tomashin**[ORCID]¤, **Francesca Fusina, Marco Marino, Alessandro Angrilli**[ORCID]*

Department of General Psychology, University of Padova, Padova, Italy

¤ Current Address: Neuroscience of Perception & Action Lab, Italian Institute of Technology, Viale Regina Elena 291, 00161 Rome, Italy
* alessandro.angrilli@unipd.it

## Abstract

Spontaneous blink is a behavioral index that proved to represent an interesting window on the human brain and mind. It can be easily extracted from the electroencephalogram, eye-tracking or video-recording. In past studies Blink Rate (BR) was found to be modulated by attention, emotion, and fatigue. We added to current research by testing some new blink-related measures (Blink Amplitude-BA, Blink Rate Variability-BRV and Blink Amplitude Variability-BAV) that could represent valid biomarkers of state and trait anxiety. We measured blinking at rest using the vertical electrooculogram extracted from the EEG recordings collected from 50 healthy female students and correlated it with their State and Trait Anxiety scores. We also correlated anxiety with the individual blinking measured during an emotional film viewing session. During resting state a good correlation among the four blink measures was found but, concerning individual differences, a significant correlation emerged only between BA and State Anxiety ($R_{48}$ = 0.288). This result is strikingly similar to that found in our past study between startle reflex amplitude and State Anxiety, and suggests that BA might represents a more ecological and easier correlate of anxiety than startle. During film viewing the correlation among blink measures was greater with respect to resting state. In addition, correlation between BA and State Anxiety was greater ($R_{44}$ = 0.557), but also BR and BRV indexes were significantly correlated with State Anxiety ($R_{44}$ = 0.418 and $R_{44}$ = −0.357 respectively). Interestingly, during clip vision the correlation between BA and Trait Anxiety became significant ($R_{44}$ = 0.354) but this effect was mediated by State Anxiety. Our results reveal how these blinking indexes are promising markers of anxiety. In particular, BA was the most effective biomarker of State Anxiety, but under specific conditions, also of Trait Anxiety. This fosters the use of blink measures in minimally invasive paradigms and experiments with ecological setting.

**Data availability statement:** All relevant data for this study are publicly available from the GitHub repository (https://github.com/alontom/Eye-Blink-Anxiety).

**Funding:** This research was funded to AA by the Italian Space Agency (ASI), Project "Space It Up!", Spoke 9, "Human Life Science & Space Medicine" - Work Package 9.4.2 "Psychological and Neurophysiological Aspects" – Contract ASI N. 2024-5-E.0 - CUP (master) I53D2400006000, and by another project funded by the Italian Ministry of Education and Research - PRIN 2022 grant, project n. 20225CKY72. MM was funded by the STARS@UNIPD program 2023 with the project entitled INTEGRATE (Inter-Network communication to Explore how simulated microGRavity can model Aging Traits on Earth). The funders had no role in study design, data collection and analysis, decision to publish, or preparation of the manuscript.

**Competing interests:** The authors have declared that no competing interests exist.

## Introduction

The search for an implicit physiological correlate for anxiety has begun more than half a century ago [1]. As a state of high vigilance, arousal and apprehension in response to an unclear potential threat, anxiety, when experienced excessively, becomes a burden on the individual's daily living [2,3,4]. Hence, the identification of an easy-to-measure objective biomarker would be helpful to assess a state of anxiety, and monitor treatment progress, by investigating the temporal evolution of this condition in different anxiety disorders. Recent research oriented to the search for anxiety biomarkers has focused on the implementation of machine learning algorithms that could analyze complex and multidimensional data, such as neuroimaging, physiological, and blood and urine profiles [5,6]. For instance, Ding et al. [7] incorporated in their investigation 16 physiological measurements derived from the electrocardiogram (ECG) and galvanic skin response (GSR) and found a 0.47 correlation coefficient between the predicted and actual state anxiety. The caveat of these approaches is the reliance on various measurements and advanced computational resources that obscure the clear biomarker-condition relationship. Indeed, there is need for a simpler and more direct psychophysiological marker of anxiety that could be easy to implement in the lab as well as in real-life experimental situations. Among the most interesting behavioral/psychophysiological markers, spontaneous blink activity and the induced blink reflex (also named startle reflex) deserve a special mention. Spontaneous blink activity was used only recently in a few behavioral investigations: among the many possible parameters, the blink rate is the most used index [8,9] and, in relation with anxiety, a very new study measured blink rate during the presentation of several emotion-inducing video clips in high vs low trait-anxiety groups [10]. Results evidenced a different emotion modulation of blink rate in high vs low trait anxiety participants.

However, in comparison with the rare and recent use of Blink Rate (BR) in psychology, the most investigated index so far for studying emotions, anxiety and psychopathology has been the blink reflex, also termed startle reflex, a quick automatic defensive reflex represented by a very fast blink induced by a strong perceptual stimulation like a thunder, a shot or a flashlight. In a past experiment using the startle reflex induced by a startling noise, during resting state, in a large sample of 111 female students [11], we found a significant correlation between State Anxiety levels and blink amplitude measured from the EMG of the *orbicularis oculi* muscle. Unexpectedly, the correlation of startle amplitude with Trait Anxiety was not significant, and we argued that in participants submitted to the lab setting (i.e., a new unfamiliar place) state anxiety tends to increase, as it is plastic and adaptive to the new situational condition, therefore is prevailing over the stable trait anxiety. Although the past experiment was theoretically relevant, the relatively low correlation (R = 0.23) and methodological complexity of the startle measure (requiring a white noise generator, headphones and synchronization between noise and EMG recording) made this psychophysiological index interesting, but not easy to use as an effective and ecological marker of anxiety.

Starting from past know-how on the significant correlation between state anxiety and startle reflex amplitude [11] and on spontaneous blink rate on both emotion and anxiety [10,8,9], with the current investigation, we aimed to propose novel anxiety biomarkers, based on spontaneous eye blinking that require a simple acquisition system and a basic processing pipeline. The rationale underlying the present study relies on the partial overlap of the neural circuits involved in startle blink reflex and spontaneous blinking, both of which include, among others, brainstem reticular formation, basal ganglia, prefrontal cortex and cerebellum [12]. The quoted neural structures have strong connections with limbic and paralimbic regions (amygdala, hippocampus, hypothalamus) involved in anxiety [13]. Startle reflex amplitude has been shown to be modulated by the amygdala and orbitofrontal cortex in patients with lesions in these brain structures [14, 15]. Thus, we hypothesized that similarly to startle reflex [11], spontaneous blinking, already demonstrated to be sensitive to anxiety [10] and emotional state [8,9], would be correlated with anxiety levels, in particular with State Anxiety levels. More in detail, we expected that blink amplitude, similarly to startle amplitude, would be positively correlated with anxiety (greater anxiety, greater blink amplitude). Furthermore, we also expected blink rate, which has been shown to be modulated by fear [9] and anxiety [10], to be positively correlated with anxiety. Eye blinking, a quick closure and reopening of the eyelid, can be monitored through different recording methods – video [10], electromyography and vertical electrooculogram- VEOG [8,9]. Treated as noise, in EEG data analysis, a massive effort has always been dedicated by researchers in human neuroscience to the identification and removal of eventual eye blinking artifacts from the EEG trace. In contrast with this general outlook, we explored the possibility of uncovering the value of these accessible and meaningful, although commonly discarded, biological data that are in the drawer of almost every human neuroscience researcher.

Eye blinking can occur voluntarily, spontaneously or as a reflex induced by intense and rapid sensory stimulation. In the latter case the blink startle reflex is elicited by external stimulation and is faster than the spontaneous endogenous one. Besides the well-known and used Blink Rate, other parameters can be extracted from a blink recording [16], including blink amplitude (BA) and the coefficient of variation of the blink amplitude (Blink Amplitude Variability; BAV). Bentivoglio et al. [17] found an average rest BR of 17 blinks per minute, while the duration of a complete average spontaneous blink ranges between 100–400 milliseconds [18]. However, similarly to other biological processes, eye blinks could also be described by the variation of their temporal dynamics, such as the coefficient of variation of the inter-blink-intervals (Blink Rate Variability; BRV), and its fractal dimension [19,20,21].

Amongst the non-invasive biomarkers, previous literature reported an association between BR and several clinical conditions. People with Schizophrenia, that are characterized by high levels of dopaminergic activity in mesolimbic areas, present higher BR compared to the general population [22]. In contrast, Parkinson's Disease patients, who exhibit an impaired dopaminergic functioning in the *substantia nigra*, show lower BR than healthy controls [23]. In particular, a closer examination suggested that the previously discovered dopamine-BR link [24,25,26] might be mediated by an indirect modulation of the spinal trigeminal complex induced by the dopamine levels in the basal ganglia [27]. Pruessner et al. [28] showed a higher dopaminergic activity in the ventral striatum during stressful tasks.

Past research investigated mainly BR during specific tasks, thus evidencing how this index is modulated by attention, arousal, emotion and fatigue [8,9]. Fewer studies are available on BR measured at rest and its correlation with individual differences such as anxiety traits. Therefore, initially, we inspected BR at rest. A resting state is beneficial since blinking occurs at a relatively slow rate (10–25 per minute) which decreases with attentional deployment, so a longer recording period is required to collect enough blinks for the analysis and avoid floor effects. Furthermore, without an external stimulus, the blink-behavior could be better attributed to internal processes and mind wandering that are associated with negative thoughts [29] and hence might be more adequate as an anxiety correlate. Other studies showed that BR increases under anxiety-inducing conditions [30,31]. In these investigations, authors manipulated the state of anxiety and focused on the intrapersonal change in blink rate during a task. However, to the best of our knowledge, no study examined the interpersonal differences in eye blinking characteristics and their association with anxiety.

In addition to the measure of eye blinking characteristics at rest, we recorded it also during a passive task during which participants viewed a stream of 18 emotional film clips, selected from the validated stimulus set "E-MOVIE" [32,33]. The used emotional clips belonged to several emotional categories of both positive and negative emotional valence. The advantage of using films instead of the more common affective pictures are two-folded. From the perspective of studying spontaneous eye blinking, a real-to-life continuous stimulus is preferred to better understand the real time dynamics as a response to a sustained affective state. Moreover, being a dynamic stimulus, videos are more ecological and evoke stronger emotional reaction than static stimuli [34].

To summarize, in the following work we investigated four parameters (BR, BA, BRV and BAV) extracted from spontaneous blinking measurements by capturing both the temporal characteristics and the magnitude of these behavioral measures. We provide a pipeline as well as a Matlab script to extract these parameters from VEOG. Then, we examined our hypothesis that 3 of these measures, especially BA, are correlated with anxiety levels. To this end, we investigated spontaneous blinking, detected via VEOG from EEG traces, both at rest and while watching a stream of emotional film clips.

## Materials and methods

### Participants

In this study, we provide a new and specific analysis of EEG data collected from a previously investigated sample described elsewhere(see for methodological details [35]). Seeking to investigate high and low emotional dysregulation levels within a community sample, 50 female healthy participants were selected out of a 422 students pool from the University of Padua. This sample is in line with sample size used in past research [9,10] and with statistical power computed for blinking measures [9]. Thus, the sample was analyzed as a unitary group but originally included two matched groups of 25 participants selected according to three questionnaires on emotional dysregulation. One group was labeled as low dysregulation and the other high dysregulation ($M_{High}$ = 22.64, $SD_{High}$ ± 2.12; $M_{Low}$ = 22.60, $SD_{Low}$ ± 1.63). Participants with past neurological disorders or receiving psychotropic treatment were excluded in the screening phase. Four participants were excluded from the film watching dataset due to partial data loss. For the aims of the present study the two groups including all healthy participants were treated as a single unitary group.

### Experimental procedure

The original experiment was approved by the Ethics Committee of the Psychology Area (protocol code 2989, date of approval 19/04/2019), University of Padova, and was conducted in line with the Declaration of Helsinki. The original experiment was aimed at studying the EEG correlates of emotion dysregulation, and results have been already published elsewhere [35]. The original sample of fifty participants was recruited from a larger sample of healthy female students, indeed participants with high pathological levels of depression or anxiety were excluded [35].On October 16th, 2023, data collected in 2019 and archived in a protected place were re-accessed for the present study, in an anonymized form (authors had no access to identification details on participants) and for a different aim and hypothesis with respect to the original one. To this end, we extracted unpublished data on anxiety levels and, from the EEG traces, we extracted blink data never analyzed from this sample so far. For the purpose of the present research aimed at studying basic properties of blink activity, any convenience sample of healthy students used in past research was acceptable, as the original hypothesis for which the sample was recruited and the conditions to which it was submitted do not affect our aims and new hypotheses. This especially holds for resting state measures but also measures collected under emotional stimulation are in line with the need to distinguish correlation of blink rate with anxiety during resting state vs during emotional stimulation condition. Furthermore, participants signed the standard informed consent suggested by our Psychology Ethics Committee for a generic use of their anonymized data, use that is not restricted to the specific hypothesis of the original experiment.

Here below the original data collection procedure is summarized.

After arriving in the lab, participants filled out and signed a written informed consent form and responded to the State Trait Anxiety Inventory (STAI, [36]) and were then connected to an EEG system (ElectroCap) with 64 tin electrodes. As a standard procedure of our EEG lab, participants with contact lenses were invited, before the recording session, to bring and use their glasses. Environmental temperature and background noise that could affect blink measures are typically controlled and stable in our dedicated psychophysiological laboratories. For the current analyses we focused on two stages of the experiment – resting state and film clips watching

First, during the 5-minutes resting state phase participants were instructed to relax sitting back on an armchair with their arms and legs uncrossed and to keep their eyes open, looking forward without a fixation cross to avoid unnecessary attentional bias. Then, participants' EEG was recorded while watching 18 different short film clips, presented on a 27 inches screen with 1280 x 720 resolution at a distance of 70 cm from participants' eyes and with sound played on stereo headphones. The clips were selected from the validated "E-MOVIE" database [32] and were presented in a randomized order. Each clip had a duration of 2 min, overall 18 clips were presented, and could belong to six emotional categories (3 negative, 2 positive and one neutral), with 3 clips per emotional category. To grant time to disengage from the emotional state raised by the previous clip, before each clip, the recording included 30 seconds of baseline, once again without a fixation cross. For the aim of the present study the blink measures were analyzed within the interval of film presentation and were all collapsed across emotional conditions within participants.

### Eye blink acquisition

Eye blinks were extracted from two tin electrodes above and below the left eye (Fp1, Ve1). The VEOG was recorded at a sampling rate of 500 Hz by a SynAmps amplifier (NeuroScan Labs, Sterling, USA), and processed with MATLAB (The MathWorks Inc.)and EEGLAB [37] while the rest of the analyses were conducted using Jamovi (The Jamovi project, 2023). To reflect the activity of the eyelid muscles, we analyzed the signals collected from the electrodes above and below each eye (Ve1, Io1, Ve2, Io2). Then, a band-pass filter (0.5–20 Hz) was applied offline to cut most artifacts [9]. Finally, we used MATLAB's built-in Find peaks function with a threshold of 100 $\mu V$ to determine a blink, keeping a minimum interval of 250ms between two consecutive blinks (Fig 1).

For the analyses below, blink amplitude (BA) was measured through the peak voltage of each blink, blink rate (BR) was measured in blinks per minute as well as temporal and magnitude variation measures – Blink Rate Variation (BRV) and Blink Amplitude Variation (BAV). To capture the temporal variation, we calculated the Coefficient of Variation (SD divided by the mean; CV) for the inter-blink-interval distribution of each segment yielding BRV. Another CV measure was computed on the amplitude distribution named BAV: both BRV and BAV accounted for the regularity of the eye blinks in magnitude and in time whereas a lower score indicates a more regular behavior. To encourage a widespread use of this approach in future research, we provide a freely available Matlab script (https://github.com/alontom/Eye-Blink-Anxiety),based on EEGLAB [37], to follow our blink acquisition pipeline from EOG signal to blink measures (Fig 2).

## Results

### Resting state session

During the resting-state condition, we found an average BR of 16.9 blinks per minute (SD = 9.28; Fig 3A), which is in line with current literature [17].The mean and dispersion of the four blink indices during resting state (left side of each panel) and during the film session (right side of each panel) are shown in Fig 3A to 3D.

First, we looked at the correlation between the four blink measures, acting as dependent variables in the next analyses, to evaluate their relationships. Table 1 indicates a low-moderate positive correlation between BR and BA, while these two indexes were negatively correlated with BRV. Additionally, the two variance measures were positively associated. These

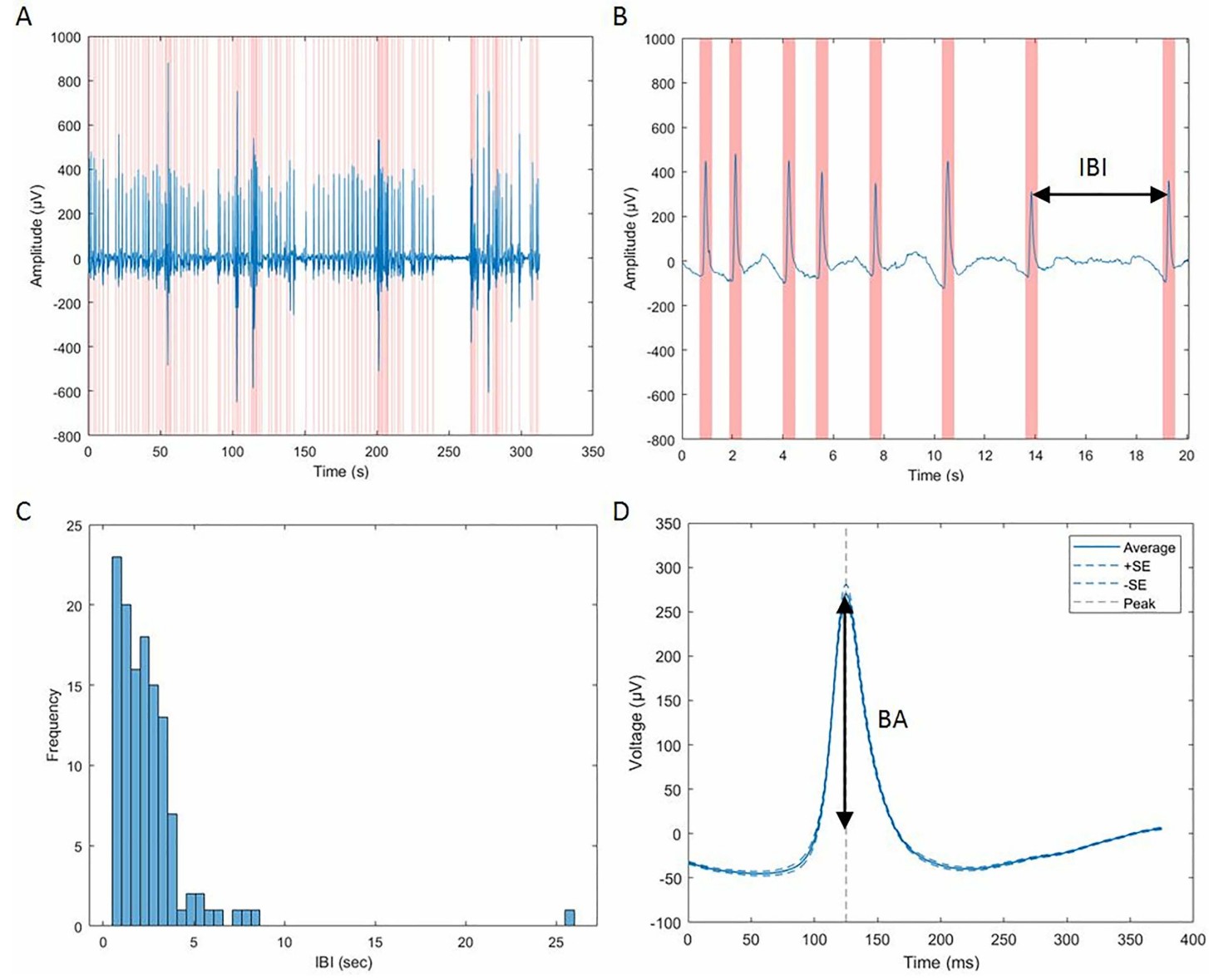

**Fig 1. A& B) Representative time series of the subtraction of the two electrodes (above and below the left eye) – on the left throughout 5 min resting state (A), and on the right a zoom-in to the first 20 seconds (B) with an arrow indicating one Inter Blink Interval (IBI).** Blinks are marked with pink areas stretching 250ms before and after the peak. C) Frequency distribution of IBI of one participant that is the basis for computing BRV. D). Average blink structure at rest with an arrow from 0 µV to the peak representing the BA.

moderate correlations may suggest that although describing the same phenomenon (spontaneous eye blinking), the two blink variance measures capture distinct aspects of it.

Assessing the relationship of the four blink measures with anxiety, we found no significant correlations with trait anxiety ($\alpha = 0.05$). However, while spontaneous eye blink rate (BR) was not associated with STAI-S above chance level ($r_{48} = 0.144$, $p = 0.319$), BA was significantly and positively correlated with STAI-S scores ($r_{48} = 0.288$, $p = 0.043$).Participants with higher state anxiety levels had greater blink amplitude (Fig 4).

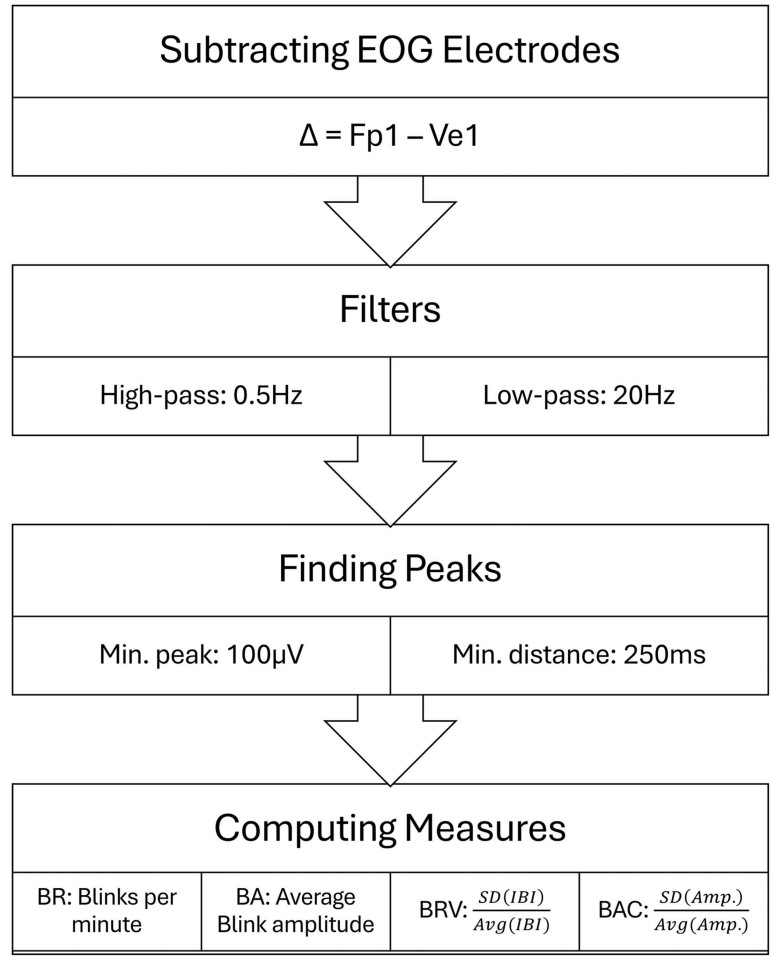

**Fig 2. A schematic representation of the 3 steps of eye blink acquisition from the VEOG signal and the quantification of blink behavior.** A Matlab script for the above process can be found on GitHub: https://github.com/alontom/Eye-Blink-Anxiety.

### Film clip session

Concerning the blink parameters during emotion-inducing films, we first examined the association between the blink measurements averaged per participant over the 18 film clips.

$$BM_{ij} = \beta_0 + u_{0j} + \beta_1 STAI_j + \varepsilon_{ij}$$

**Equation 1.** A mixed model predicting participant j's blink measurement (BM) when watching film i. $\beta_0$: overall intercept, $u_{0j}$: random intercept for each participant, $\beta_1$: fixed effect of STAI and $\varepsilon_{ij}$: residual error (unexplained variance)

$$BM_{ij} = \beta_0 + u_{0j} + \beta_1 STAI_j + \beta_2 Type_{ij} + \beta_3 (Type_{ij} \cdot STAI_j) + \varepsilon_{ij}.$$

**Equation 2.** A mixed model predicting participant j's blink measurement (BM) when watching film *i*. $\beta_0$: overall intercept, $u_{0j}$: random intercept for each participant, $\beta_1$: fixed effect of STAI, $\beta_2$: fixed effect of film type effect, $\beta_3$: interaction effect between Type and STAI and $\varepsilon_{ij}$: residual error (unexplained variance).

We computed the association between the four blink indexes and anxiety levels. For this purpose, mixed linear models predicting blink measurements (BR, BA, BRV and BAV) were conducted to account for the

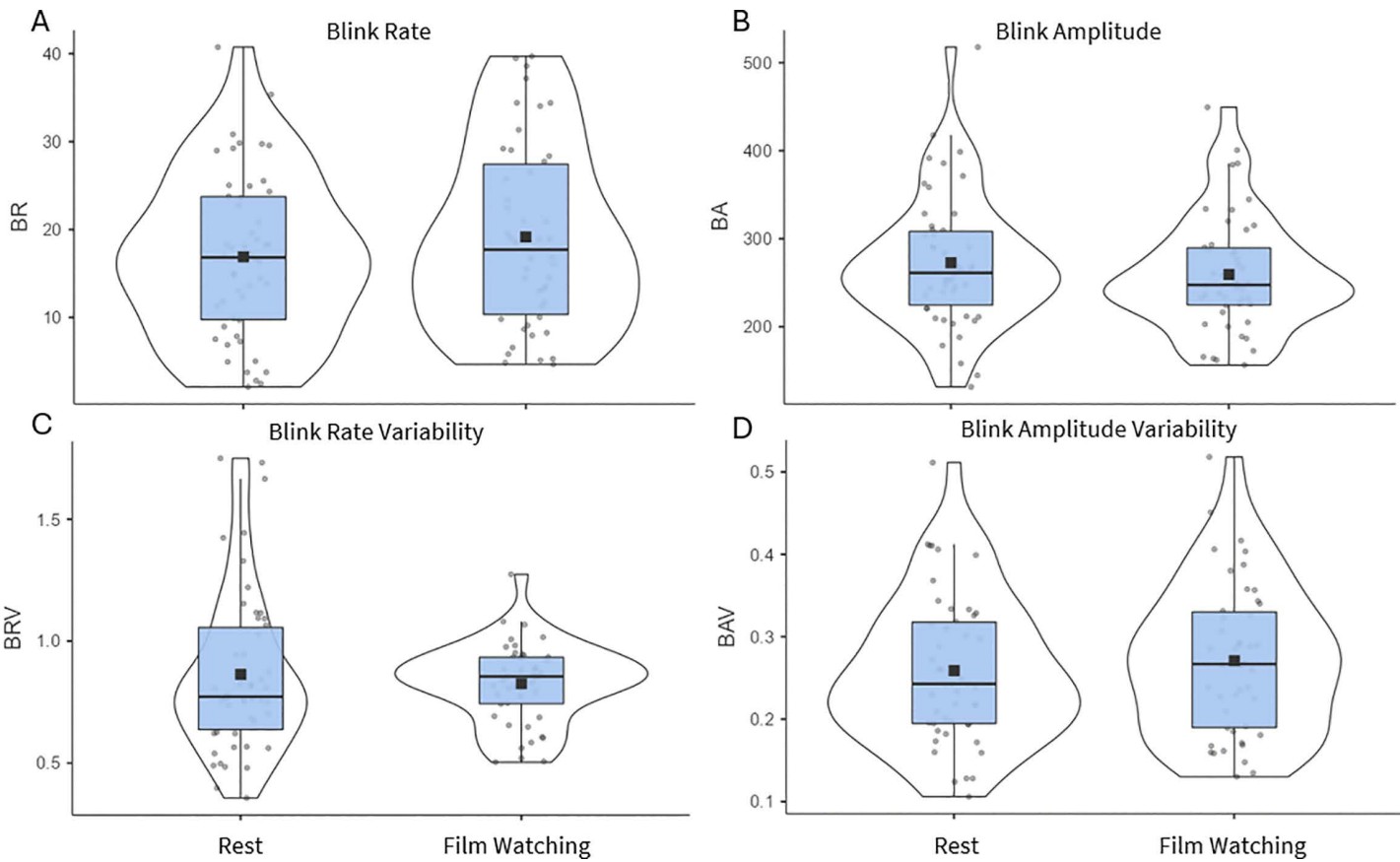

**Fig 3. Violin plots representing the distributions of the four eye blinking measures.**

**Table 1. Pearson's correlations (df = 48) among the four blink measurements and STAI-S as well as STAI-Tat rest. Significant results are in bold.**

**Correlation Matrix**

|  |  | BR | BRV | BA | BAV | STAI-S | STAI-T |
|---|---|---|---|---|---|---|---|
| BR | Pearson's r | — |  |  |  |  |  |
|  | p-value | — |  |  |  |  |  |
| BRV | Pearson's r | **−0.311** | — |  |  |  |  |
|  | p-value | 0.028 | — |  |  |  |  |
| BA | Pearson's r | **0.352** | **−0.350** | — |  |  |  |
|  | p-value | 0.012 | 0.013 | — |  |  |  |
| BAV | Pearson's r | −0.147 | **0.414** | −0.045 | — |  |  |
|  | p-value | 0.309 | 0.003 | 0.759 | — |  |  |
| STAI-S | Pearson's r | 0.144 | −0.226 | **0.288** | −0.236 | — |  |
|  | p-value | 0.319 | 0.115 | 0.043 | 0.098 | — |  |
| STAI-T | Pearson's r | 0.001 | −0.175 | 0.215 | −0.268 | **0.670** | — |
|  | p-value | 0.993 | 0.224 | 0.134 | 0.060 | <.001 |  |

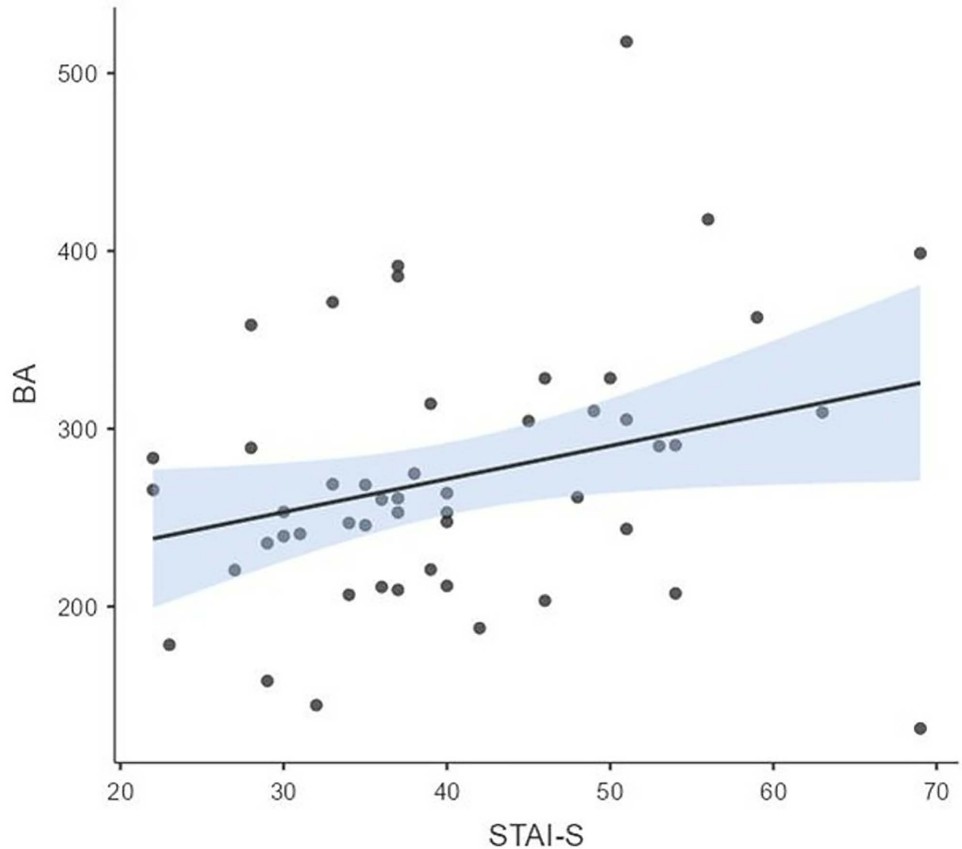

**Fig 4. The association between STAI-S and blink amplitude in the resting state session ($R_{48}$ = 0.288).**

relationship between the 18 observations of each individual using subject as a clustering variable (random intercept)addressing the statistical dependency among repeated observations within each subject, and the STAI score as a predictor (see Equation 1). First, Trait anxiety significantly predicted only BA with $R^2_{marginal}$ = 9.88% ($\beta_1 : F_{1,44}$ = 6.32, $p$ = 0.016).

Concerning the influence of clip category, a mixed model predicting participant blink characteristics (A – BR, B – BA, C–BRV) based on STAI-S and film type was used (see Equation 2). No difference was found among correlations of anxiety and blink characteristics with the different film types as all intercepts were overlapped ($\beta_3 : F_{5,771}$ < 1.19, $p$ > 0.32; see Appendix Figure),thus film type was not included in the following analyses. Additionally, significant effects of film type were found on all blink characteristics ($\beta_2 : F_{5,771}$ > 10.067, $p$ < 0.001.

Interestingly, when inspecting current levels of anxiety, STAI-S predicted 24.5% of the variance in BA ($\beta_1 : F_{1,44}$ = 19.8, $p$ < 0.001). In addition, BR and BRV were predicted by STAI-S ($\beta_1 : F_{1,44}$ = 9.32, $p$ = 0.004; $\beta_1 : F_{1,44}$ = 6.12, $p$ = 0.017). Importantly, these effects remained significant when averaging BA, BR or BRV scores across film clips for each participant (Table 2). However, while BR was positively associated with STAI-S, subjects with higher STAI-S showed lower BRV, meaning lower variability in their inter blink interval.

The results presented in Table 2 are similar to those reported for resting state (Table 1), but the extent of correlation was, on average, larger. In particular, the correlations among BR, BRV and BA were greater compared to the resting state condition, and a slightly higher correlation was found between the two variability measures, i.e., BRV and BAV.

**Table 2. Pearson's correlations (df = 44) between the blink measurements, STAI-S and STAI-T during the film presentation session; all the 18 clips were averaged within each participant. Significant results are in bold.**

**Correlation Matrix**

|  |  | BR | BA | BRV | BAV | STAI-S | STAI-T |
|---|---|---|---|---|---|---|---|
| BR | Pearson's r | — |  |  |  |  |  |
|  | p-value | — |  |  |  |  |  |
| BA | Pearson's r | **0.574** | — |  |  |  |  |
|  | p-value | <.001 | — |  |  |  |  |
| BRV | Pearson's r | **−0.694** | **−0.547** | — |  |  |  |
|  | p-value | <.001 | <.001 | — |  |  |  |
| BAV | Pearson's r | **−0.297** | −0.009 | **0.477** | — |  |  |
|  | p-value | 0.045 | 0.951 | <.001 | — |  |  |
| STAI-S | Pearson's r | **0.418** | **0.557** | **−0.349** | −0.072 | — |  |
|  | p-value | 0.004 | <.001 | 0.017 | 0.634 | — |  |
| STAI-T | Pearson's r | 0.178 | **0.354** | −0.121 | −0.025 | **0.686** | — |
|  | p-value | 0.236 | 0.016 | 0.422 | 0.868 | <.001 | — |

A further analysis was carried out to detail the two relationships of trait and state anxiety with blink amplitude since a significant association between STAI-T and STAI-S was observed, alongside a weaker correlation between BA and STAI-T, and a stronger one with STAI-S (Fig 5). The employed model revealed the mediation role of state anxiety in the correlation between trait anxiety and BA. Our analysis on the BA (averaged across films) yielded a significant indirect effect ($Z = 3.085$, $p = 0.002$), while the direct effect in this model, STAI-T on BA, was not significant ($Z = −0.311$, $p = 0.756$).

## Discussion

In this work, we aimed to investigate the potential of four eye-blinking measures as possible biomarkers of anxiety levels across individuals. To test the above hypotheses, we identified the relationships between four measures derived from spontaneous blinking BR, BA, BRV and BAV and state- as well as trait-anxiety. Furthermore, the novel use of all these blink parameters, analyzed together for the first time, allowed us to also measure the correlations among all these indices.

Starting from the correlation among measures, there was evidence that all four indices were moderately correlated with each other, both during resting state and during film clip watching. This points to a common source of variability (i.e., blinking phenomenon) for all four measures but also to a relative independence and specificity of these indices. In detail, the two main measures BR and BA showed a moderate correlation during resting state ($r = 0.352$) that became larger during film session ($r = 0.574$): indeed, those participants showing a higher number of blinks per minute showed also a faster closure (i.e., amplitude) of the eyelid. We also found a negative association between both BR and BA with BRV ($r < −0.300$)

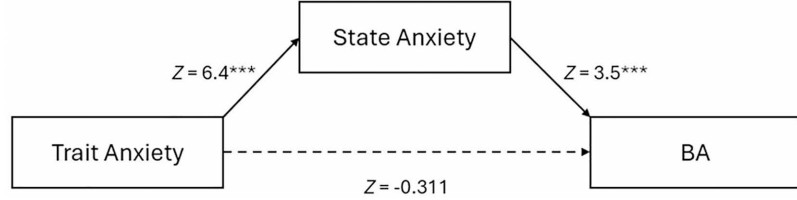

**Fig 5. A model predicting BA from trait anxiety (STAI-T) via the mediation of state anxiety (STAI-S).** *** (p < 0.001).

that was larger during film session (r < −0.500). This should be carefully interpreted since the BRV formula includes the average inter blink interval that is the inverse of BR. Based on this term, the correlation between BRV and BR is supposed to be positive, suggesting that the unexpected negative overall association found is derived from a negative link between BR and the standard deviation of the inter blink interval. Interestingly, the two variability measures, BRV and BAV, were positively correlated during both resting state and film session (r > 0.400).

From another view, we can also consider the effect sizes of the links among our variables: indeed Person's correlation, in addition to the statistical significance which depends on sample size, is also an effect size, with r = 0.1–0.2 representing a small effect, r = 0.3 a medium effect, and r = 0.5 a large effect [38]. Thus, most of our effects ranged from medium (r = 0.288) to medium-large (r = 0.352) to large (r = 0.5 or higher).This provides further insight on the strength of our effects.

Concerning the correlation between blink measures and anxiety, our main hypothesis started from a past study [11] showing a significant correlation of startle reflex amplitude with State Anxiety, but not with Trait Anxiety, in a sample of 111 female students. We expected that the new measures derived from spontaneous blinking would have shown similar effects as compared with startle amplitude. More in detail, among the four indices of the present work, we found that BA was the measure closest to startle reflex amplitude, thus we expected that especially BA would have shown a positive correlation with state anxiety. What are the differences and commonalities between the two measures? Startle reflex is induced by a strong rapid sensory stimulation and is measured by the intensity/amplitude of the integrated EMG of the *orbicularis oculi* muscle, the main drive of eyelid closure. Spontaneous blink amplitude instead is a large potential produced by the electrostatic effect of the shift of the eyelids over the *cornea* during eye closure: the faster is this shift the larger is the voltage measured as BA. The two measures, startle reflex and BA, are necessarily similar for three reasons. First, blink amplitude measured by the integrated EMG (representing the overall spectral energy of muscle contraction) of the *Orbicularis Oculi* muscle is expected to be correlated with another measure of the same effector, such as, the eyelid shift leading to a VEOG signal proportional to the speed of eye closure. This may be analogue to the measure of the heart rate by means of the electrocardiogram versus the photopletismogram: two methods measuring the same psychophysiological parameter (i.e., heart rate). Second, the two measures share the same neural network underlying blink generation(see the most recent review: [12]), which includes the brainstem, basal ganglia, the trigeminal nerve, but of course they are not equivalent because one is a very fast defense blink reflex elicited by strong external stimulation (e.g., a shot or a flash or lightning), the other is a slow spontaneous and erratic blink controlled by internal cerebral processes indirectly modulated by external not locked stimulation. Third, we expected that anxiety level would have been positively correlated with both blink amplitude and blink rate as one important hub modulating blink activity is the brainstem, more precisely the reticular formation also termed ARAS (Ascending Reticular Activating System) important for modulating arousal at cortical level through the diffused serotoninergic system. One of the main features of anxiety is the hyper-arousal leading to insomnia, tiredness, jittery. Fear eliciting stimuli as well as anxiety, activate the amygdala, which in turn, through its central nucleus, activates the reticular formation, and this leads to both a potentiated startle reflex [14,39] and higher blink rate [9]. In the present study we did not seek to demonstrate that the two methods are more or less correlated, instead we aimed to show that, in line with our hypotheses based on the known neural circuitry of the two measures, we expected an interesting convergence between the two blink amplitude methods in their capacity to correlate with state anxiety during both resting state and emotional stimulation conditions. Indeed, the correlation found here between BA and STAI-S (r = 0.288) was very close to that found in past study [11] between startle amplitude and STAI-S (r = 0.23). Similarly to startle reflex, the correlation of BA with trait anxiety STAI-T, during resting state, was not significant. This again points to the relevance of the moderate situational-state anxiety induced by a novel place, such as a lab setting on participants rather than to the influence of trait anxiety on blink measures. The convergence between BA and startle points to a strong similarity between the two measures when used for assessing anxiety, with the advantage that they can be used in different experimental paradigms and settings: startle reflex can be used for short or static stimuli, is more invasive (i.e., it is elicited by strong sound shot)and suited for lab experimental settings, while blink amplitude and blink rate are more suited

for ecological not-interfering or real-to-life studies during relatively long recording/stimulation lasting at least 30–60 sec. A further novel result of the present study was the greater correlation between BA and STAI-S reached during the film session (r = 0.557). This also applied to the other two blink indexes, BR and BRV, whose positive and negative (respectively) correlations with state anxiety increased and became significant. Concerning BR, the possible correlation with anxiety was expected starting from the observation in past studies that blink rate in emotional settings was modulated by arousal levels [9] and was increased when anxiety was induced in lab [10,30,31].

Of special interest is the correlation between BA and trait anxiety that during the film session increased and reached the statistical significance: this was the only blink index which was correlated with STAI-T. Results suggested that overall BA is a stronger correlate of anxiety in comparison with the other three blink indices as it was found in both resting state and during film session and was correlated also with trait anxiety during film session.

In summary, our investigation posits that higher state anxiety is characterized, depending on the setting, by at least one of the following: higher blink rate, stronger blinks (higher BA) and a more regular blinking pattern (lower BRV).

On top of that, our mediation model built on results from the film session (see Fig 5) indicated an indirect relationship between STAI-T and BA with STAI-S as an intervening variable with no significant direct effect of STAI-T on BA. In other words, trait anxiety predicts state anxiety which then predicts BA. This finding suggests that blink characteristics at a certain time point represent the current affective state of an individual to a higher degree than their stable trait does. Thus, BA shares more information with a situational experience (STAI-S) than with a fixed trait (STAI-T). Besides that, depending on the circumstances, trait anxiety, or individual's predisposition to respond, would shape their individual experience of anxiety or distress [40].

In the present study we aimed to contrast two important conditions used in the literature typically in different paradigms and contexts. Having in one session both resting state and visual stimulation allowed us to infer how the environment in two important ecological conditions influence blinking. Indeed one condition is passive characterized by mind wandering self-reflection and stimulus rejection, the other condition is the opposite, being characterized by stimulus intake and blinking tracked by specific stimuli grabbing attention. In general, we found a stronger association between blinking characteristics and anxiety during the film clip watching than during the resting state condition. This is consistent with past study [10] (on blink rate only) reporting an interaction between anxiety and emotion during the presentation of emotional videoclips. This finding raises a question regarding the need for a visual stimulus to constrain eye blinking for a better representation of internal anxiety states. Indeed, we expected resting-state blink characteristics to be highly associated with state anxiety in agreement with studies discussing the phenomenon of rest related negative affect (RRNA; [41]), which links resting state, or being off task, condition with negative thoughts as well as less happiness [42,32]. Our results contradict this hypothesis. Therefore, two mechanisms, both valid and probably additive, could explain the correlation difference found between resting state and film session. First, although each film clip was shorter, about 2 min, compared to the resting state condition, the 18 films overall resulted in 36 min of blink recording, thus providing a more stable and reliable blink measure with respect to the 5-min resting state recording. A second mechanism would be the reduction in unexplained interpersonal variance due to the focus on the visual channel and the repetition of similar visual stimuli. In the rest condition, without a focus on the external world, a broad range of erratic thoughts and emotions may occur by increasing variance between participants that is unrelated to the previously reported state of anxiety. Indeed, visual stimuli could narrow down the range of thoughts and, by limiting mind wandering, allowed a better representation of interpersonal differences in spontaneous blinking. As a further consideration, the visual stimulus characteristics might also have played a role in increasing the correlation between blink measures and anxiety: the film clips were a mix of several emotional categories, pleasant, neutral, and unpleasant. Although taken individually the BR of each film clip did not correlate with anxiety measures, when collapsing data from all 18 clips a significant correlation emerged. The lack of the correlation between anxiety and blink rate on a single clip basis is probably due to the use of only one anxiety measure at the beginning of the recording session and since state anxiety level is expected to change across the recording session, especially

soon after aversive/fearful film clips, this single measure may explain an overall effect over the whole film session but not specific pointwise film-anxiety correlations.

One limit of our study is that the sample population consisted only of female students half of whom had high scores in emotional dysregulation levels, hence our results might be not generalized to a male population. Furthermore, the high dysregulation group showed higher STAI-S than the low dysregulation group (Welch's $t_{41.4} = -3.6$, $p < 0.001$). This difference probably contributed to increase the effect sizes, however in further analysis, no significant effects were found between the groups in blink measures. It is also possible that sample selection based on emotional dysregulation increases overall noise and variances because of the use of not-homogenous samples in traits other than anxiety.

## Conclusions

Our aim in this work was to demonstrate how new blink measures are correlated with anxiety in two different settings, resting state and a film session. Results should encourage researchers, with an emphasis on EEG focused laboratories, to incorporate spontaneous blinking measurements in their studies as readily available, simple-to-analyze, yet insightful index of mental states. While BR and BRV can be easily recorded also by means of eye-trackers and video-recording, BA and BAV can be more easily measured by means of the VEOG trace from EEG recordings. However, BA (and related BAV) could be in principle extracted from video-recording performed at high speed and high resolution by measuring the speed of eye closure, an index probably highly correlated with VEOG: this new method deserves to be further investigated in future studies. In general, given the non-invasiveness of blink measures, these allow to avoid interference with the mental processes under investigation, e.g., during interaction of the participant with other people, with the environment, with many stimuli, in non-cooperative individuals such as newborns, and in individuals at distance such as in online video transmissions. Taken together, blink measured through the induced startle reflex (there are thousands of studies with this method) or by the recording of spontaneous blinking in ecological experiments, showed an impressive potential for future research due to its ability to reveal many psychological domains of human mind: attention, emotion, fatigue, lie detection, anxiety, personality, psychiatric and neurological disorders. The correlation we found between state anxiety and BA in resting state, very similar to that found between startle reflex amplitude and state anxiety [11] provides an important external and convergent validity of the novel blink measures. In the current work, we showed that, when measured in specific contexts (e.g., emotion-inducing films), various eye blinking measurements are potent, non-invasive indices for state and trait anxiety also between individuals. BA and BR during film clip watching were strongly correlated with STAI-S ($r = 0.557$, $r = 0.418$, large and medium-large effect sizes respectively) in a degree comparable to well-established physiological correlates of anxiety such as skin conductance and heart rate variability measures usually showing up to $|r| = 0.5$ [7,43]. Specifically, this study highlights the value of the novel BA as an anxiety biomarker. Further research on larger samples and with different characteristics (gender, age, neurological as well as psychiatric pathologies) promises to disclose a new branch of psychobiological ecological measures.

## Supporting information

**S1 Fig.** Mixed models predicting participant blink characteristics (A – BR, B – BA, C- BRV) based on STAI-S and film type. No difference was found among correlations of anxiety with the different film clips as all intercepts were overlapped, this justifies considering the movie session as a block in which measures collected from different clips were collapsed within participants.
(TIF)

## Author contributions

**Conceptualization:** Alessandro Angrilli.

**Data curation:** Alon Tomashin.

**Formal analysis:** Alon Tomashin.

**Funding acquisition:** Alessandro Angrilli.

**Investigation:** Francesca Fusina, Marco Marino.

**Methodology:** Alon Tomashin, Francesca Fusina, Marco Marino, Alessandro Angrilli.

**Project administration:** Alessandro Angrilli.

**Resources:** Francesca Fusina.

**Software:** Alon Tomashin.

**Supervision:** Alessandro Angrilli.

**Visualization:** Alon Tomashin.

**Writing – original draft:** Alon Tomashin.

**Writing – review & editing:** Francesca Fusina, Marco Marino, Alessandro Angrilli.

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
