## [Decision Letter · Decision Letter 0]

25 Jun 2025

Dear Dr. Angrilli,

Thank you for submitting your manuscript to PLOS ONE. After careful consideration, we feel that it has merit but does not fully meet PLOS ONE’s publication criteria as it currently stands. Therefore, we invite you to submit a revised version of the manuscript that addresses the points raised during the review process.

We look forward to receiving your revised manuscript.

Kind regards,

Nick Fogt

Academic Editor

PLOS ONE

Journal Requirements:

“This research was funded to AA by the Italian Space Agency (ASI), Project “Space It Up!”, Spoke 9, "Human Life Science & Space Medicine" - Work Package 9.4.2 "Psychological and Neurophysiological Aspects" – Contract ASI N. 2024-5-E.0 - CUP (master) I53D2400006000, and by another project funded by the Italian Ministry of Education and Research  - PRIN 2022 grant, project n. 20225CKY72.

MM was funded by the STARS@UNIPD program 2024 with the project entitled INTEGRATE (Inter-Network communication to Explore how simulated microGRavity can model Aging Traits on Earth).”

Additional Editor Comments:

There are significant concerns raised by both reviewers regarding this manuscript. Of high significance, reviewer #1 points out that the mechanism/hypothesis for a correlation between spontaneous blinks and anxiety is not well established in the paper. The reviewer has provided a recently published paper as a reference that perhaps applies in this regard. Reviewer #1 also points out the potentially concerning issue that this study looks at those data gathered previously for another purpose. One question that arises is whether the methodology would have been different had the authors originally considered the question that is now being addressed in the manuscript. Additionally, the reviewer raises the very significant question of whether the activities described in the manuscript are properly covered under the original human subjects protocol/consent form.

Reviewer #2 raises some concerns that may be difficult to address. Specifically, the reviewer comments on the "lack of a control group and demographic diversity" as well as the "modest sample size". Please address all of the reviewer comments.

Reviewers' comments:

Reviewer's Responses to Questions

**Comments to the Author**

1. Is the manuscript technically sound, and do the data support the conclusions?

Reviewer #1: No

Reviewer #2: Partly

2. Has the statistical analysis been performed appropriately and rigorously?

Reviewer #1: Yes

Reviewer #2: I Don't Know

3. Have the authors made all data underlying the findings in their manuscript fully available?

Reviewer #1: No

Reviewer #2: Yes

4. Is the manuscript presented in an intelligible fashion and written in standard English?

Reviewer #1: No

Reviewer #2: Yes

Reviewer #1: Summary:

In this manuscript, the authors extract characteristics of the spontaneous eye blink (blink rate, blink rate variability, blink amplitude, and blink amplitude variability) from previously recorded EEG signals and correlate these characteristics to participants’ trait and state anxiety ratings. The authors find that spontaneous blink amplitude at rest positively correlates with state, but not trait anxiety. They also find that blink rate while watching video clips positively correlates to state anxiety, while blink rate variability is negatively correlated. Again, blink amplitude is positively correlated to state anxiety and, under these conditions, also to trait anxiety. The authors note that state anxiety mediates the relationship between trait anxiety and blink amplitude. The authors argue that these correlations justify using spontaneous blink characteristics, especially blink amplitude, as a biomarker for anxiety.

While I don’t doubt the correlations the authors have found, I do question the value of the findings and the validity of the applications they envision. Below, I note that the main limitation of this manuscript is a failure to fully establish the theory of why the spontaneous blink would correlate to measures of anxiety. In addition, I do have concerns about the methodology, which was done post hoc. Because the authors did not originally intend to measure the spontaneous eyeblink, they did follow the typical protocols for doing so. Without replication, the correlation of one blink characteristic to state anxiety feels tenuous.

Background/Theory:

1. The authors review past literature (their own) showing that the amplitude of a reflexive, startle blink correlates with state but not trait anxiety measures. Given the nature of their paradigm, which used an acoustic startle stimulus, this relationship makes sense. However, the authors then argue that the overlap between the circuitry of the spontaneous blink and the startle blink predicts that the spontaneous blink should show the same relationship with measures of anxiety. In addition, given the relative ease of measuring the spontaneous blink, they note that it might be ideal compared to the reflexive blink and may even be more “strongly sensitive” to emotional state compared to the startle blink.

I find this argument, as it is currently presented, fairly weak. While there is overlap in these circuits, the spontaneous blink is far more removed from circuitry related to anxiety regulation than the autonomic response to startle stimuli. It would be helpful if the authors detailed the specific relationship between anxiety and the spontaneous blink circuit to demonstrate why they would expect a correlation and why it might be more strongly sensitive to emotional state as they do.

2. The authors state, “However, to the best of our knowledge, no study examined the interpersonal differences in eye blinking characteristics and their association with anxiety.” They should be aware that a study was published in March titled “ Emotional Blink Patterns: A Possible Biomarker for Anxiety Detection in a HCI Framework” (https://doi.org/10.1007/s42979-025-03810-y). I encourage the authors to incorporate this study into their literature review and interpretation.

3. I am confused about why the authors did not make directional predictions about the expected correlation between the characteristics of the blink. It seems that if the theory was there to link the spontaneous blink with measures of anxiety, the authors would have expectations about the specific ways the blink would be correlated to these measures.

Methodology:

1. I have some concerns that this study was not planned a priori, but rather the data were accessed 3 years after they were collected and used to assess a second question. In addition to ethical concerns (e.g., does this study fall under the language of informed consent participants agreed to?), it also means the study wasn’t developed with the measurement of spontaneous eye blink in mind. For example, when collecting the spontaneous blink, it is standard to ask participants to remove contact lenses because those can impact the characteristics of the spontaneous blink. Was this done in this sample? In addition, because factors that affect the spontaneous blink (e.g. dopamine level) are sensitive to environmental conditions such as room temperature, these are usually controlled for. This should, at the very least, be noted as a limitation of the study.

2. I am not clear on the purpose or value of recording the blink during the emotional video clips, since the researchers ended up averaging across different emotions (including neutral). I agree with their assessment that this likely provided a more stable reflection of the blink, given the longer timeline. However, we know that tasks change the blink parameters, and the authors do not address this at all. Ultimately, it’s not clear how this part of the study helps us to understand the relationship between the spontaneous blink and anxiety, aside from showing that we see the amplitude relationship in both resting and task states.

Discussion:

1. Similar to the introduction, I find the interpretation of why the spontaneous blink would be related to measures of anxiety to be fairly weak. The authors state: “We expected that the new measures derived from spontaneous blinking would have shown similar or even stronger effects as compared with startle amplitude.” and I’m not clear why it would be a stronger effect. Again, it makes sense that the amplitude of a startle blink would correlate to state anxiety, but not why the amplitude of the spontaneous blink would. This is especially true since the authors note that the mechanism is not equivalent (despite them talking about the similarity of the overlapping circuits).

2. In their conclusion, the authors argue that this could be an easy, non-invasive proxy of anxiety in participants, such as in “individuals at distance such as in telematic video transmissions.” However, they also note that “While BR and BRV can be easily recorded also by means of eye-trackers and video-recording, BA and BAV can be more easily measured by means of the VEOG trace from EEG recordings." If it’s only the blink amplitude that is informative about anxiety and blink amplitude needs some type of EEG/EOG measure, then that doesn’t seem that it can be used in the manner they are suggesting.

Writing Style:

1. I encourage an in-depth review of this manuscript by someone whose native language is English. There are many grammatical errors and typos (e.g. the use of “addictive” rather than what I assume is “additive”, inconsistent use of past tense, use of “less” rather than “fewer”, etc.)

2. I suggest a reorganization of the flow of the introduction. For example, it would make sense to present the basic definitions of the spontaneous blink and the startle blink before the literature on how these might relate to anxiety.

Reviewer #2: This paper investigates the use of blink parameters—especially Blink Amplitude (BA), Blink Rate (BR), and their variability—as possible psychobiological markers of anxiety. Using vertical electrooculogram (VEOG) extracted from EEG recordings, the authors aim to correlate these blink indices with both State and Trait Anxiety, assessed during rest and while watching emotion-inducing films.

Following are some observations:

1.Since eye movements can be captured with ease and non-invasively in naturalistic conditions, blink-metrics certainly offer real world application potential. BA and BR and their variations can also act as potential candidates for biomarkers

2. The experiment seems to have considered a sample having high and low ED in the absence of any healthy control. The correlation of BA with state-anxiety will be similar for healthy control also. It will also be similar for rest state with less value. However, during emotional film-clip viewing due to high or low arousal the BA variation will be more. But, again, this is true for healthy control also. Hence, can a threshold be established to distinguish between healthy control and anxiety affected with BA variation. Yes, it is true for State anxiety, the BA variation certainly counts. But the correlation found out is so less (<0.5) and the sample consists of only female candidates. BA will also vary from emotion to emotion individually. In emotional film viewing, how far it is convincing to take the average of BA variation. Rather, BA variation should be studied separately for +ve and -ve emotions.

But when it comes to Trait Anxiety, which is more stable and less sensitive to immediate stimuli, claiming a strong association from a modest correlation (-0.35) in a small, homogeneous sample feels premature.

3. BR and BRV can be more significant candidates as biomarkers for exploration of trait anxiety and for distinguishing between healthy control and anxiety affected. Again, these should be studied separately for +ve and -ve emotions.

4. Eqation1) and 2) should be clearly explained along with significance of the coefficients.

5. can it be justified in quantitative terms: “trait anxiety is mediated by state anxiety” with BA variation and BA as parameters?

Summary of my observations are as follows:

• The resting state correlations tell us more about the inter-relatedness of blink metrics than anxiety per se.

• The findings during emotional stimulation seem more valid for State Anxiety, but even there, the statistical strengths aren't that sound.

• For Trait Anxiety, the associations are speculative at best without a larger, diverse sample and a more rigorous analytical design.

• The study has novelty and the level of enthusiasm is commendable, but some methodological gaps like lack of a control group and demographic diversity, modest sample size, and speculative generalizations limit the strength of its claims

**Do you want your identity to be public for this peer review?** For information about this choice, including consent withdrawal, please see our Privacy Policy

Reviewer #1: No

Reviewer #2: No

---

## [Author Response · Author response to Decision Letter 1]

7 Oct 2025

REBUTTAL LETTER

(see also the file "Response to Reviewers" as the text is formatted for a better reading with different characters)

PONE-D-25-09086

The eye, a spy hole on human mind: spontaneous blink rate and amplitude, and their variability, as new psychobiological markers of anxiety

Reviewer # 1 Q1 - While I don’t doubt the correlations the authors have found, I do question the value of the findings and the validity of the applications they envision. Below, I note that the main limitation of this manuscript is a failure to fully establish the theory of why the spontaneous blink would correlate to measures of anxiety. In addition, I do have concerns about the methodology, which was done post hoc. Because the authors did not originally intend to measure the spontaneous eyeblink, they did follow the typical protocols for doing so. Without replication, the correlation of one blink characteristic to state anxiety feels tenuous.

Authors - even if we have revised the ms. to improve its readability, we do not agree with the reviewer for several reasons: 1) we have explained the rationale of our study, starting from blink(startle) reflex-anxiety relationship and the literature, both in the intro and in the discussion. Now in the revision we have further explained 2) why we expected blink rate and amplitude to be correlated with state anxiety explaining how the neural hub modulated by anxiety for both startle and spontaneous blink is the reticular system (ARAS), 3) we have cited the previous paper showing a BR-anxiety relationship suggested the reviewer (it was published in the same month during which we submitted our paper, so we could not know it), 4) we had established the direction of the relationship and now this is more explicitly written, so that it should be clear that our study was hypothesis-driven (see also other points) and not explorative.

Concerning the methodology: ours were not post hoc analyses, we had completely new hypotheses and we carried out new analyses on a sample of data collected for different hypotheses and aims. This is exactly what data science is doing by re-analyzing past data from public repositories, that is, by reformulating new hypotheses on data collected in the past and reusable for different aims and hypotheses.

Concerning "without replication..the correlation....feels tenuous", most of the scientific literature does not involve replicated results but original results: without the publication of the original study is not possible to replicate it. Statistics serve to demonstrate that results are not random. We further clarified this issue at pg. 7 see below:

"For the purpose of the present research aimed at studying basic properties of blink activity, any convenience sample of healthy students used in past research was acceptable, as the original hypothesis for which the sample was recruited and the conditions to which it was submitted do not affect our aims and new hypotheses. This especially holds for resting state measures but also measures collected under emotional stimulation are in line with the need to distinguish correlation of blink rate with anxiety during resting state vs during emotional stimulation condition."

In addition, since the reviewer claimed that our results were "fairly weak" several times and we do not understand where there is this evidence as we discussed only significant statistics, we decided to report also the well known statistical concept that Pearson's correlation, besides significance, is also an index of the Effect Size and our effect sizes can be considered medium to large for most correlations (see below, and pg.12 highlighted in yellow):

"From another view, we can also consider the effect sizes of the links among our variables: indeed Person's correlation, in addition to the statistical significance which depends on sample size, is also an effect size, with r= 0.1-0.2 representing a small effect, r = 0.3 a medium effect, and r = 0.5 a large effect [26]. Thus, most of our effects ranged from medium (r= 0.288) to medium-large (r= 0.352) to large (r= 0.5 or higher).This provides further insight on the strength of our effects."

Reviewer # 1 Q2 - The authors review past literature (their own) showing that the amplitude of a reflexive, startle blink correlates with state but not trait anxiety measures. Given the nature of their paradigm, which used an acoustic startle stimulus, this relationship makes sense. However, the authors then argue that the overlap between the circuitry of the spontaneous blink and the startle blink predicts that the spontaneous blink should show the same relationship with measures of anxiety. In addition, given the relative ease of measuring the spontaneous blink, they note that it might be ideal compared to the reflexive blink and may even be more “strongly sensitive” to emotional state compared to the startle blink.

I find this argument, as it is currently presented, fairly weak. While there is overlap in these circuits, the spontaneous blink is far more removed from circuitry related to anxiety regulation than the autonomic response to startle stimuli. It would be helpful if the authors detailed the specific relationship between anxiety and the spontaneous blink circuit to demonstrate why they would expect a correlation and why it might be more strongly sensitive to emotional state as they do.

Authors - we agree with the reviewer that we had no reasons to expect that blink measure would be more sensitive and strong in the relationship with anxiety than startle, so we have canceled this both in the introduction and discussion. Concerning reviewer's observation that our argument on common circuitry of blink reflex (startle) and spontaneous blinking is fairly weak we do not agree, so we have detailed and made more compelling our assumptions on how anxiety may affect the two measures (spontaneous blinking and startle), in almost one page of the discussion, and along three lines of evidence (see below and text highlighted pg. 13):

"The two measures, startle reflex and BA, are necessarily similar for three reasons. First, blink amplitude measured by the integrated EMG (representing the overall spectral energy of muscle contraction) of the Orbicularis Oculi muscle is expected to be correlated with another measure of the same effector, such as, the eyelid shift leading to a VEOG signal proportional to the speed of eye closure. This may be analogue to the measure of the heart rate by means of the electrocardiogram versus the photopletismogram: two methods measuring the same psychophysiological parameter (i.e., heart rate). Second, the two measures share the same neural network underlying blink generation (see the most recent review: [7]), which includes the brainstem, basal ganglia, the trigeminal nerve, but of course they are not equivalent because one is a very fast defense blink reflex elicited by strong external stimulation (e.g., a shot or a flash or lightning), the other is a slow spontaneous and erratic blink controlled by internal cerebral processes indirectly modulated by external not locked stimulation. Third, we expected that anxiety level would have been positively correlated with both blink amplitude and blink rate as one important hub modulating blink activity is the brainstem, more precisely the reticular formation also termed ARAS (Ascending Reticular Activating System) important for modulating arousal at cortical level through the diffused serotoninergic system. One of the main features of anxiety is the hyper-arousal leading to insomnia, tiredness, jittery. Fear eliciting stimuli as well as anxiety, activate the amygdala, which in turn, through its central nucleus, activates the reticular formation, and this leads to both a potentiated startle reflex [1, 5] and higher blink rate [31]. In the present study we did not seek to demonstrate that the two methods are more or less correlated, instead we aimed to show that, in line with our hypotheses based on the known neural circuitry of the two measures, we expected an interesting convergence between the two blink amplitude methods in their capacity to correlate with state anxiety during both resting state and emotional stimulation conditions."

Reviewer # 1 Q3 - The authors state, “However, to the best of our knowledge, no study examined the interpersonal differences in eye blinking characteristics and their association with anxiety.” They should be aware that a study was published in March titled “ Emotional Blink Patterns: A Possible Biomarker for Anxiety Detection in a HCI Framework” (https://doi.org/10.1007/s42979-025-03810-y). I encourage the authors to incorporate this study into their literature review and interpretation.

Authors - as already written above, we have quoted this paper several times in both the introduction and the discussion (ref. # 9).

Reviewer # 1 Q4 - I am confused about why the authors did not make directional predictions about the expected correlation between the characteristics of the blink. It seems that if the theory was there to link the spontaneous blink with measures of anxiety, the authors would have expectations about the specific ways the blink would be correlated to these measures.

Authors - we thank the reviewer as we forgot to specify explicitly in what direction we expected the correlation of blink measures with anxiety. This could be implicitly inferred from startle reflex amplitude that was increased in high anxiety individuals, so we expected that blink amplitude and blink rate (known to be modulated by arousal levels) were positively correlated with anxiety levels. We have added the terms "positively correlated" in both the intro and the discussion sections. We have also added a sentence in the intro (pg.4) clarifying our predictions:

"More in detail, we expected that blink amplitude, similarly to startle amplitude, would be positively correlated with anxiety (greater anxiety, greater blink amplitude). Furthermore, we also expected blink rate, which has been shown to be modulated by fear [31] and anxiety [9], to be positively correlated with anxiety."

Reviewer # 1 Q5 - I have some concerns that this study was not planned a priori, but rather the data were accessed 3 years after they were collected and used to assess a second question. In addition to ethical concerns (e.g., does this study fall under the language of informed consent participants agreed to?), it also means the study wasn’t developed with the measurement of spontaneous eye blink in mind. For example, when collecting the spontaneous blink, it is standard to ask participants to remove contact lenses because those can impact the characteristics of the spontaneous blink. Was this done in this sample? In addition, because factors that affect the spontaneous blink (e.g. dopamine level) are sensitive to environmental conditions such as room temperature, these are usually controlled for. This should, at the very least, be noted as a limitation of the study.

Authors - concerning the concept of "study a-priori planned" we have already answered this question, the study was a-priori planned and the new measures and analyses were a-priori planned three years after the data collection for completely different aims and hypotheses from the original setting and acquisition, therefore this was a new study, and the raw data should be considered a convenience sample from which it was possible to extract the new measures (see also above and highlighted parts pg. 7 of the Ms.). We have also clarified that our informed consent (the standard form is written by the Ethics Committee) does not restrict the use of the data for one specific aim and study, see below (pg. 7).

"Furthermore, participants signed the standard informed consent suggested by our Psychology Ethics Committee for a generic use of their anonymized data, use that is not restricted to the specific hypothesis of the original experiment."

Concerning the procedure, it is important also for a good EEG recording to avoid contact lenses so we have added this sentence (pg.7):

"As a standard procedure of our EEG lab, participants with contact lenses were invited, before the recording session, to bring and use their glasses. Environmental temperature and background noise that could affect blink measures are typically controlled and stable in our dedicated psychophysiological laboratories."

Reviewer # 1 Q6 - I am not clear on the purpose or value of recording the blink during the emotional video clips, since the researchers ended up averaging across different emotions (including neutral). I agree with their assessment that this likely provided a more stable reflection of the blink, given the longer timeline. However, we know that tasks change the blink parameters, and the authors do not address this at all. Ultimately, it’s not clear how this part of the study helps us to understand the relationship between the spontaneous blink and anxiety, aside from showing that we see the amplitude relationship in both resting and task states.

Authors - we think we had addressed this issue in our discussion. Now, in the revision, we have further extended in several points the discussion on this issue. We better explained the rationale, we have also quoted the suggested paper on anxiety during emotional video-stimulation (ref. #9), see below (and pg. 15):

"In the present study we aimed to contrast two important conditions used in the literature typically in different experiments and contexts. Having in one session both resting state and visual stimulation allowed us to infer how the environment in two important ecological conditions influence blinking. Indeed one condition is passive characterized by mind wandering and self-reflection and stimulus rejection, the other condition is the opposite, being characterized by stimulus intake and blinking tracked by specific stimuli grabbing attention. In general, we found a stronger association between blinking characteristics and anxiety during the film clip watching than during the resting state condition. This is consistent with past study [9] (on blink rate only) reporting an interaction between anxiety and emotion during the presentation of emotional videoclips ".

Furthermore, we remind that in the original version of our manuscript we also tried to explain the difference observed in the two conditions by suggesting two possible mechanisms, (see bottom of pg.15):

"First, although each film clip was shorter, about 2 min, compared to the resting state, condition the 18 films overall resulted in 36 min of blink recording, thus providing a more stable and reliable blink measure with respect to the 5-min resting state recording. A second mechanism would be the reduction in unexplained interpersonal variance due to the focus on the visual channel and the repetition of similar visual stimuli. In the rest condition, without a focus on the external world, a broad range of erratic thoughts and emotions may occur by increasing variance between participants that is unrelated to the previously reported state of anxiety. Indeed, visual stimuli could narrow down the range of thoughts and, by limiting mind wandering, allowed a better representation of interpersonal differences in spontaneous blinking."

Reviewer # 1 Q7 - Similar to the introduction, I find the interpretation of why the spontaneous blink would be related to measures of anxiety to be fairly weak. The authors state: “We expected that the new measures derived from spontaneous blinking would have shown similar or even stronger effects as compared with startle amplitude.” and I’m not clear why it would be a stronger effect. Again, it makes sense that the amplitude of a startle blink would correlate to state anxiety, but not why the amplitude of the spontaneous blink would. This is especially true since the authors note that the mechanism is not equivalent (despite them talking about the similarity of the overlapping circuits).

Authors - we have already addressed this issue in our answer to Q1 and Q2, we have canceled the adverb "strongly" in both the introduction and discussion. Again see our response to Q2 and read Ms. pg.13 for a detailed explanation of the neural circuitry and common hub explaining how anxiety can modulate both

---

## [Decision Letter · Decision Letter 1]

6 Nov 2025

Dear Dr. Angrilli,

Thank you for submitting your manuscript to PLOS ONE. After careful consideration, we feel that it has merit but does not fully meet PLOS ONE’s publication criteria as it currently stands. Therefore, we invite you to submit a revised version of the manuscript that addresses the points raised during the review process.

We look forward to receiving your revised manuscript.

Kind regards,

Nick Fogt

Academic Editor

PLOS ONE

Journal Requirements:

Additional Editor Comments:

Thank you for your thorough and detailed responses to the reviewer comments. The reviewer asks for clarification on "the inclusion of the random intercept when overall intercept (beta0) is already present". I believe the explanation for this is in the manuscript in that the random intercepts address differences for each subject from the overall intercept. Is that correct, and do you think this needs further clarification? Please briefly address this by adding a sentence explaining what the difference in overall and random intercepts is, if you believe this is appropriate.

Reviewers' comments:

Reviewer's Responses to Questions

**Comments to the Author**

Reviewer #2: All comments have been addressed

2. Is the manuscript technically sound, and do the data support the conclusions?

Reviewer #2: Yes

3. Has the statistical analysis been performed appropriately and rigorously?

Reviewer #2: Yes

4. Have the authors made all data underlying the findings in their manuscript fully available?

Reviewer #2: Yes

5. Is the manuscript presented in an intelligible fashion and written in standard English?

Reviewer #2: Yes

Reviewer #2: The eye, a spy hole on human mind: spontaneous blink rate and amplitude, and their variability, as new psychobiological markers of anxiety: The title of the paper suits well with it's subject matter and appeals to students and researchers focused on noninvasive approaches to study mental health. Several eye parameters have been useful in the study of human emotions. Studies to identify biomarkers of affective states carries significance in the area of mental health research. However, limited available resources significantly hinder mental health research efforts. I appreciate the efforts of the authors to clarify the questions raised and justify their claims.

One suggestion as follows may be included in the manuscript: "explanation on the inclusion of the random intercept when overall intercept (beta0) is already present" for clarity.

The paper can be accepted.

**Do you want your identity to be public for this peer review?** For information about this choice, including consent withdrawal, please see our Privacy Policy

Reviewer #2: No

---

## [Author Response · Author response to Decision Letter 2]

19 Nov 2025

Reviewer #2: The eye, a spy hole on human mind: spontaneous blink rate and amplitude, and their variability, as new psychobiological markers of anxiety: The title of the paper suits well with it's subject matter and appeals to students and researchers focused on noninvasive approaches to study mental health. Several eye parameters have been useful in the study of human emotions. Studies to identify biomarkers of affective states carries significance in the area of mental health research. However, limited available resources significantly hinder mental health research efforts. I appreciate the efforts of the authors to clarify the questions raised and justify their claims.

One suggestion as follows may be included in the manuscript: "explanation on the inclusion of the random intercept when overall intercept (beta0) is already present" for clarity.

The paper can be accepted.

Authors: we thank the reviewer for acknowledging our efforts to improve clarity of our Ms. As suggested we have added a sentence at pg. 10 (highlighted in capital letters) to clarify the "... inclusion of the random intercept...".

“For this purpose, mixed linear models predicting blink measurements (BR, BA, BRV and BAV) were conducted to account for the relationship between the 18 observations of each individual using subject as a clustering variable (random intercept) ADDRESSING THE STATISTICAL DEPENDENCY AMONG REPEATED OBSERVATIONS WITHIN EACH SUBJECT, and the STAI score as a predictor (see Equation 1).”

---

## [Editor Report · Decision Letter 2]

20 Nov 2025

The eye, a spy hole on human mind: spontaneous blink rate and amplitude, and their variability, as new psychobiological markers of anxiety

PONE-D-25-09086R2

Dear Dr. Angrilli,

We’re pleased to inform you that your manuscript has been judged scientifically suitable for publication and will be formally accepted for publication once it meets all outstanding technical requirements.

Kind regards,

Nick Fogt

Academic Editor

PLOS ONE

Additional Editor Comments (optional):

Thank you again for addressing all of the reviewer and editor comments.
---

## [Editor Report · Acceptance letter]

28 Nov 2025

PONE-D-25-09086R2

PLOS ONE

Dear Dr. Angrilli,

I'm pleased to inform you that your manuscript has been deemed suitable for publication in PLOS ONE. Congratulations! Your manuscript is now being handed over to our production team.

Kind regards,

on behalf of

Dr. Nick Fogt

Academic Editor

PLOS ONE